# Catechol-O-methyltransferase and dopamine receptor D4 gene variants: Possible association with substance abuse in Bangladeshi male

**Jahanara Akter Sonia[1], Tohfa Kabir[2], M. M. Towhidul Islam[3], Yearul Kabir[3]***

**1** Department of Biochemistry and Molecular Biology, Gono Bishwabidyalay, Savar, Bangladesh,
**2** Department of Biochemistry, Primeasia University, Dhaka, Bangladesh, **3** Department of Biochemistry and Molecular Biology, University of Dhaka, Dhaka, Bangladesh

* ykabir@yahoo.com

**Data Availability Statement:** All relevant data are within the manuscript and its Supporting Information files.

## Abstract

Genetic risk of substance abuse is encoded mainly by central neurochemical pathways (mostly dopaminergic system) related to reinforcement and reward. In this study a functionalpolymorphism in Catechol-O-methyltransferase (COMT) (Val158Met) and the Dopamine receptor D4 gene (DRD4) (120 bp tandem duplication) has been studied in substance abused subjects. The study was carried out with 183 substance abused subjects and 175 healthy persons with no history of substance abuse. DNA was extracted and polymorphisms were analyzed using allele-specific PCR. The impact of these two polymorphisms was also analyzed on addictive characteristics (age of starting abuse, a pattern of drug habit, and period of addiction). It was found that only the heterozygous variant of COMT polymorphism (Val/Met) (p<0.05, OR = 1.66, 95% CI = 1.044–2.658) and both homozygous (p<0.05, OR = 0.43, 95% CI = 0.193–0.937) and heterozygous (p<0.05, OR = 0.37, 95% CI = 0.172–0.826) derived variants of DRD4 120 bp tandem duplication were significantly associated with risk of substance abuse compared to controls. In case of association of these polymorphisms with an age of onset, no significant difference was found among three different genotypic groups of COMT polymorphism. Whereas, the homozygous derived variant (240 bp/240 bp) of DRD4 gene was found to have a later age of onset (20.5±0.8) for substance abuse compared to heterozygous (120 bp/240 bp) (19.1±0.8) and wild type homozygous variant (120 bp/120 bp) (16.0±0.5), which was statistically significant (p<0.05). Again, in the case of the pattern of drug habit, the frequency of the Val/Val genotype is higher in polysubstance abused (>2 drugs) subjects (p<0.05) compared to the heterozygous Val/Met containing variants. An association of period of addiction was analyzed with an individual type of substance abuse and found that heroin abused subjects have a significantly higher period of addiction (11.6±1.0) compared to other abusers (p<0.01). Further, it was found that Met/Met containing variants of COMT polymorphism has a more extended period of addiction than other genetic variants in heroin abused subjects. These results indicate that genetic variability may influence the susceptibility to the risk of substance abuse and addictive characteristics.

**Funding:** The first author gratefully acknowledges to the Ministry of Science and Technology, Government of the People's Republic of Bangladesh for providing financial support in her research through NST fellowship.

**Competing interests:** The authors have declared that no competing interests exist.

## Introduction

The geographic location and availability of drugs made Bangladesh as one of the worst victims of the drug. According to the "Annual Drug Report of Bangladesh, 2016" substance abuse is a national concern. It is now prevalent everywhere in Bangladesh and millions of people suffer from the problem [1]. In a previous study, epidemiological data shows that the number of substance abusers has increased rapidly over recent years and a very high percentage (93%) of addicted subjects in Bangladesh are male [2]. The influence of both genetic and environmental factors is responsible for substance abuse, a complex neurodegenerative disorder [3,4]. The major factors include socioeconomic disparities, physical and mental health, social integration/isolation, and overall quality of life. Again, about half of someone's risk of addiction is embedded in their genes [5]. Individuals become abused to substances like alcohols, opioids, amphetamines, etc. through involving themselves in different stages of addiction that includes: experimentation/anticipation, then light or moderate use to heavy use and finally to a stage of risky use which will lead to substance dependence and abuse [6].

Long term administration of addictive drugs results in hampering neurotransmitters' normal function by producing amplified messages [7], and this repetitive exposure may promote compulsive drug-seeking behaviors [8]. As substance abuse is a disorder in the brain reward system, the natural reward activities are influenced by artificial stimuli of addictive drugs. Interestingly, neuroimaging studies have confirmed the limbic system's involvement and the role of dopamine in the rewarding process and thereby in substance dependence [9–11]. Therefore, abnormality in genes encoding proteins related to dopamine processing in the brain can be responsible for causing addiction.

The chromosomal location of the Catechol-O-methyltransferase (COMT) gene is 22q11 which is expressed in dopaminergic brain regions and regulates dopamine level as it encodes a dopamine metabolizing enzyme [12,13]. COMT contains a functional codon 158 polymorphism situated in exon 4 of chromosome 22, which encodes either a valine (GTG) or a methionine (ATG). It has been found that the homozygous valine genotype containing enzyme activity is 40% higher than homozygous methionine containing enzyme and the heterozygous variants have an intermediate level of activity [9,13]. Thus, due to decreased metabolism, the methionine variant carriers have more dopamine in their prefrontal cortex, which may be responsible for many of the neuropsychological associations [14]. In the African population, it was found that Val158Met polymorphism of the COMT gene has a significant association with cocaine dependence [15]. Association with COMT was also found in methamphetamine abused individuals in the Taiwanese population [4]. The Dopamine D4 receptor (DRD4) gene VNTR polymorphisms have also been association with substance abuse in several studies [4,16,17]. DRD4 encodes D4 receptors that are members of D2 like G-protein-coupled receptor family [18]. The DRD4 gene contains several polymorphic sites including 48-bp tandem repeat in exon 3 [17] and 120-bp tandem duplication at 1.2 kb upstream from the initiation codon [19,20]. As the 120bp tandem duplication region contains consensus sequences for different transcription factors, the involvement of that polymorphism to protein expression has been hypothesized where the shorter allele is having higher transcriptional activity than the longer allele [4,21].

Early age onset is associated with developing subsequent drug-related problems and increased duration of addiction [22]. However, in different studies, it was found that the age of onset can have significant influences on different genotypic groups of genetic polymorphism [23,24]. Again, the pattern of drug habit may also play an influential role in substance abuse. It is found that the effect of polymorphism is more frequent in polysubstance abuse [25] and a high frequency of homozygous valine variant of COMT polymorphism was found in polysubstance abuse [26].

Therefore, in this study, along with the investigation of the frequency of COMT Val158Met and DRD4 120bp VNTR polymorphisms in substance abuse, we have also discussed the influence of these polymorphisms on the age of onset, pattern of drug habit and period of addiction in Bangladeshi substance abusers. The present study hypothesized that the carrier of the derived allele of both COMT Val158Met and DRD4 120 tandem duplication polymorphisms in the dopaminergic system are associated with the risk of substance abuse.

## Materials and methods

### Subject

This study was carried out with 183 male substance-dependent patients (case) belonging to the age group of 15–49 years and 175 control male subjects belonging to the same age group. The patients were recruited from the "Central Drug Addiction Treatment Center" Tejgaon, Dhaka, where they were under proper treatment and care. The control subjects were recruited from the Department of Biochemistry and Molecular Biology, University of Dhaka, and two hospitals of Dhaka city while they came for a regular checkup. Eligibility criteria for addicted subjects included age at least 15 years or older; they must be dependent on drugs not less than 1 year with no history of major physical disorders. Healthy volunteers with no history of substance abuse were recruited as control. All the experiment subjects were male, as we collected samples from a rehabilitation center where only male addicted subjects were admitted. Further, due to social stigma, female addicted subjects do not usually come for treatments in the rehabilitation centers. All study subjects completed a structured questionnaire by researcher taking face-to-face interviews covering information on socio-demographic characteristics, e.g., age, height, weight, residential, occupational and smoking history, addiction history, addiction types, and family history of addiction. Written consent was taken from all study subjects, and in the case of participants under the age of 18, written consent was taken from their legal guardian. The study was approved by the Institutional Ethical Review Committee (BMBDU-ERC/EC/18/016) of the Department of Biochemistry and Molecular Biology, University of Dhaka, and conducted following the declaration of Helsinki and its subsequent revisions [27].

### Sample collection

Approximately five (5.0) ml of venous blood was collected from each individual with an expert nurse's help and transferred into a sterile tube containing ethylenediaminetetraacetic acid disodium (EDTA-Na$_2$). Blood samples were kept in an ice chamber following collection and during transportation. The blood samples were stored at –20˚C until used.

### DNA extraction and genotyping

Genomic DNA was extracted from lymphocytes of peripheral blood collected in EDTA, using a standard phenol-chloroform method, followed by ethanol precipitation according to the protocol used by Bailes et al. [28] and Hosen et al. [29]. The concentration and purity of extracted DNA were determined by NanoDrop spectrophotometer at 260 and 280 nm. Candidate gene analysis was carried out by Polymerase Chain Reaction (PCR) method in a DNA thermal cycler (Applied Biosystem, USA). The PCR reaction was carried out according to the method described by Hoda et al. [30] and Seaman et al. [20], respectively, for COMT Val 158Met and DRD4 120bp VNTR polymorphisms. All PCR products were separated by electrophoresis in an adequate percentage of agarose gel, stained with ethidium bromide, and visualized under UV light. Figs 1 and 2 represent the COMT and DRD4 gene PCR products.

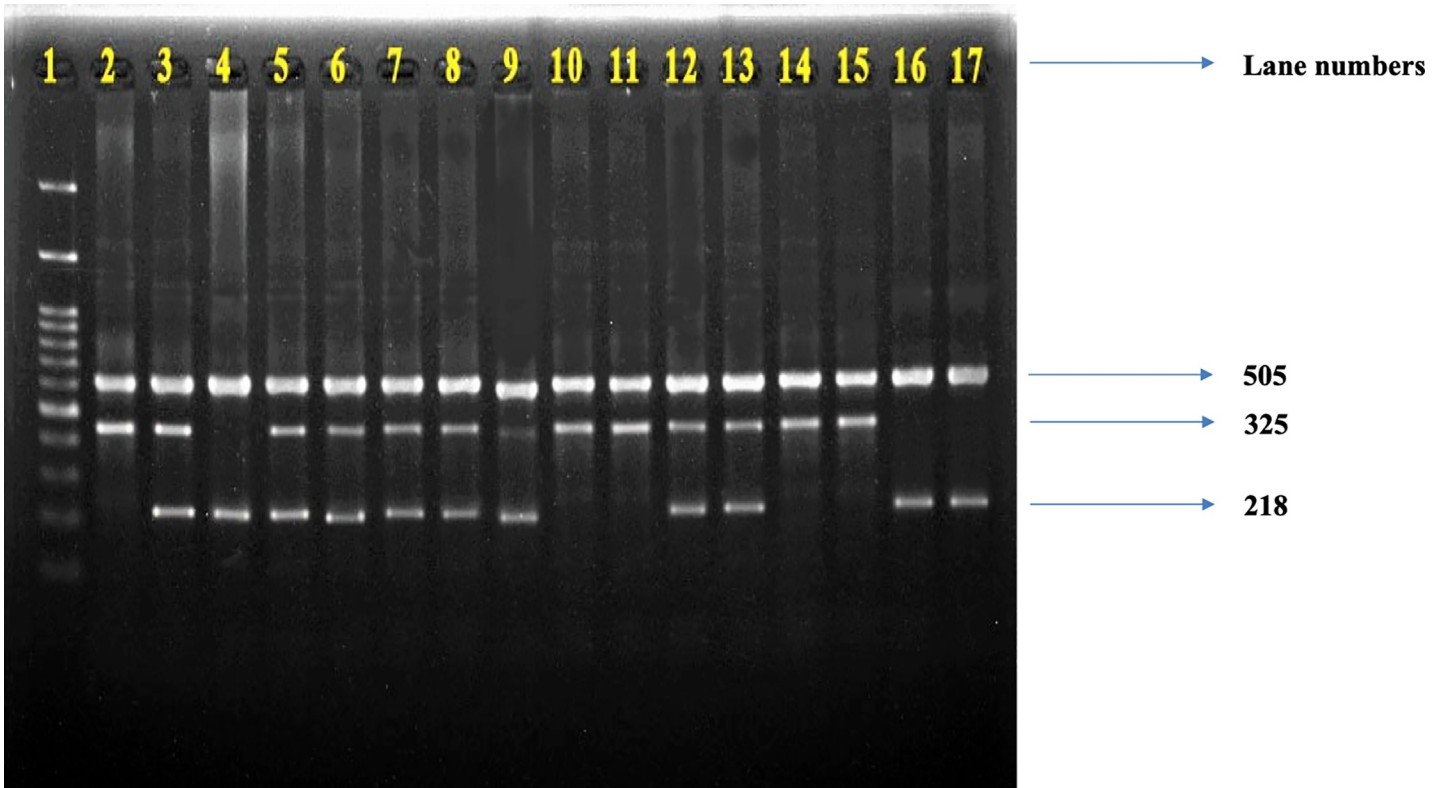

**Fig 1. Representative picture of COMT gene PCR products.** The presence of 505 bp and 325 bp fragments on lanes 1, 9, 10, 13, 14 indicates the existence of homozygous Val/Val (G/G) allele, the presence of three fragments of 505 bp, 325 bp and 218 bp on lanes 2, 4, 5, 6, 7, 8, 11, 12 indicates the existence of heterozygous Val/Met (G/A) allele, while the presence of 505 bp and 218 bp on lanes 3, 15 16 indicates the homozygous Met/Met (A/A) allele. Lane 1 indicates a 100 bp DNA ladder (Thermo Fisher Scientific, Invitrogen DNA ladder 15628019).

## Statistical analysis

Data were expressed with a percentage in the categorical variable and with mean ± SEM in case of calculating numerical data. The analyses were carried out using GraphPad Prism (version 7.0). In the case of genotypic analysis, Yates' continuity corrected- or Pearson's chi-square test were performed when observed frequencies were less or more than 25, respectively. The contingency table was used to compare categorical values to controls, and odd ratios (OR) from Chi-square were used as a risk measure at 95% confidence intervals (95% CI). Multivariate logistic regression was performed to estimate the risk of demographic characteristics among the genotypes. As our sampling procedure followed a normal distribution, unpaired t-test (two-tailed) and one-way ANOVA were performed to analyze the association of age of onset and period of addiction of substance abuse with different COMT and DRD4 genotypes. After One-way ANOVA, Tukey's test for Post-Hoc analysis was also done to compare pairwise differences. Differences were considered significant with $p < 0.05$.

## Results

### Baseline characterization of the study population

The baseline characteristics of the substance abused and control subjects are shown in Table 1. In this study, the majority (79.2%) of drug abusers belong to age above 20 years. Again, a significantly higher percentage of drug abusers are less educated (84.7%) and unemployed (39.9%) compared to the control subjects. Multiple logistic regression showed that in the case

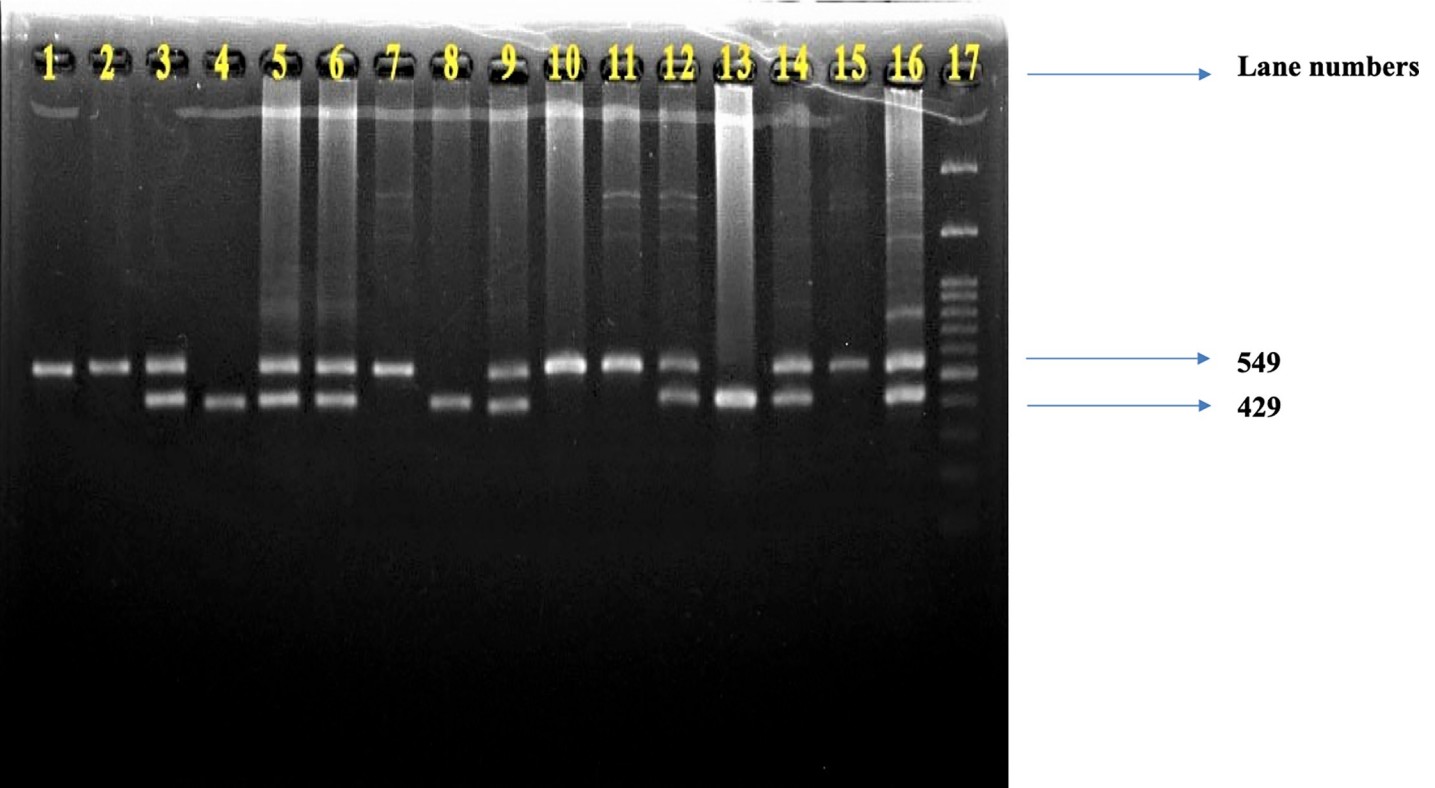

**Fig 2. Representative picture of DRD4 gene PCR products.** The presence of both 549 bp and 429 bp fragments on lanes 3, 5, 6, 9, 12, 14, 16 indicates the existence of heterozygous 120 bp/240 bp allele, the presence of only 549 bp fragment on lanes 1, 2, 7, 10, 11, 15 indicates the existence of homozygous 240 bp/240 bp allele, while the presence of only 429 bp on lanes 4, 8, 13 indicates the homozygous 120 bp/120 bp allele. Lane 17 indicates a 100 bp DNA ladder.

of COMT polymorphism, the graduate and employed subjects are less likely to be addicted than the secondary educated and unemployed subjects.

The mean body mass index (BMI) values were 20.6±0.3 (kg/m$^2$) and 24.8±0.4 (kg/m$^2$) in drug-addicted and control subjects respectively. After multiple logistic regression, it was found that BMI showed a significant association, which reveals that for 1 unit increase in BMI, there will be 18% less chance to be addicted. In the case of smoking status, drug-addicted subjects were 100% smoker than 34.9% in control subjects.

### Behavioral and clinical characteristics of substance abused subject

As shown in Table 2, the behavioral pattern of drug abusers indicated that the highest percentage of abusers take drugs through smoking (39.8%), most of them were polysubstance abusers (43.2%) with a moderate magnitude of intake (58.0%), peer pressure is the main reason of addiction among abusers (48.9%), a minimum percentage of the abusers showed relapse behavior (9.5%) and suicidal attempt (5.9%).

### Association analysis of COMT Val158Met and DRD4 120bp VNTR polymorphism

The genotype frequencies for the COMT Val158Met and DRD4 120bp VNTR polymorphism in the substance abused and control subjects are shown in Table 3. A significant difference was found in genotype frequencies of COMT Val158Met, but no significant difference was found

**Table 1. Baseline characteristics of the study subject.**

| Variable | Substance Abuser (n = 183) | Control (n = 175) | *p*-value | |
|---|---|---|---|---|
| **Age (Year)**[†] | 28.3±0.6 | 28.6±0.8 | ns | |
| ≤ 20 | 38 (20.8) | 23 (13.1) | | |
| > 20 | 145 (79.2) | 152 (86.9) | ns | |
| **Education Status** | | | | |
| Secondary | 155 (84.7) | 67 (38.3) | <0.001 | |
| Graduation | 28 (15.3) | 108 (61.7) | | |
| **Employment Status** | | | | |
| Employed | 110 (60.1) | 146 (83.4) | <0.001 | |
| Unemployed | 73 (39.9) | 29 (16.6) | | |
| **Blood Pressure** (mm Hg)[†] | | | | |
| Systolic | 110.6±1.8 | 114.0±1.1 | ns | |
| Diastolic | 73.4±1.4 | 77.3±0.7 | ns | |
| **BMI** (kg/m²)[†] | 20.6±0.3 | 24.8±0.4 | <0.001 | |
| **Smoking Status** | | | | |
| Smoker | 183 (100) | 61 (34.9) | <0.001 | |
| Non-Smoker | 0 (0) | 114 (65.1) | | |

Results expressed as number (percentage)

[†]Mean±SEM; BMI: Body Mass Index; p<0.05 was considered as the level of significance; ns: not significant.

in allele frequencies (OR = 1.29, 95% CI = 0.959–1.752) compared to the control group. Here, heterozygous mutant Val/Met genotype containing group has 1.66 (OR = 1.66, 95% CI = 1.044–2.658) times higher risk of substance abuse than the control subjects. Likewise, in DRD4 120 bp VNTR polymorphism, there were significant differences in genotype frequencies between substance abusers and controls, but no significant difference was found in allele frequencies (OR = 0.81, 95% CI = 0.593–1.110). Both heterozygous (120bp/240bp) (OR = 0.37, 95% CI = 0.172–0.826) and homozygous (240 bp/240 bp) (OR = 0.43, 95% CI = 0.193–0.937) derived variants were found to show protective role for substance abuse compared to the control subjects which was statistically significant.

Further, all the addicted subjects were categorized into three groups- methamphetamine, heroin, and cannabis abusers to analyze whether this COMT and DRD4 polymorphisms have an association with any specific drug. Here, in case of categorizing the substance abused subjects into a particular group, they were divided according to their drug-taking habit (Table 4). To be more specific, if an individual has taken a specific drug (e.g., methamphetamine) regularly and more frequently for more than two years recently, he was categorized as this specific drug abuser (e.g., methamphetamine abuser). Surprisingly, it was found that COMT polymorphism has a significant influence on the only methamphetamine abused subjects who carried heterozygous derived variants (p<0.05). In the case of DRD4 polymorphism, it was found that both the heterozygous (120bp/240bp) and homozygous(240bp/240bp) derived variants have less risk to become cannabis abusers which was statistically significant (p<0.01).

## Frequency distribution of COMT and DRD4 polymorphism according to the age of onset for substance abuse

In this study, among the substance abused subjects the earliest age of onset taking drugs was at 10 years old, and the oldest was 44 years old. These subjects started abusing drugs mostly at an early age of adulthood (19.3±0.5 years, mean±SEM). In the case of the COMT Val158Met

**Table 2. Behavioral and clinical characteristics of substance abused subject.**

|  | 1 | 2 | 3 | 4 | 5 | Missing Values |  |
|---|---|---|---|---|---|---|---|
| **Route of Administration** | 70 (39.8) | 36 (20.5) | 12 (6.8) | 9 (5.1) | 49 (27.8) | 7 | 1. Smoke |
|  |  |  |  |  |  |  | 2. Swallow |
|  |  |  |  |  |  |  | 3. Snort |
|  |  |  |  |  |  |  | 4. Injection |
|  |  |  |  |  |  |  | 5. Multiple |
| **Pattern of Addiction** | 42 (23.8) | 58 (33.0) | 76 (43.2) |  |  | 7 | 1. Single |
|  |  |  |  |  |  |  | 2. Double |
|  |  |  |  |  |  |  | 3. Multiple |
| **Type of Drug Dependence** | 38 (21.6) | 46 (26.1) | 23 (13.1) | 13 (7.4) | 56 (31.8) | 7 | 1. Methamphetamine* |
|  |  |  |  |  |  |  | 2. Cannabis* |
|  |  |  |  |  |  |  | 3. Heroin* |
|  |  |  |  |  |  |  | 4. Others§ |
|  |  |  |  |  |  |  | 5. Multiple† |
| **Cause of Addiction** | 87 (48.9) | 39 (21.9) | 12 (6.7) | 33 (18.5) | 7 (3.9) | 5 | 1. Peer Influence |
|  |  |  |  |  |  |  | 2. Interest |
|  |  |  |  |  |  |  | 3. Family Problem |
|  |  |  |  |  |  |  | 4. Depression |
|  |  |  |  |  |  |  | 5. Multiple and Others |
| **Relapse Behavior** | 16 (9.5) | 153 (90.5) |  |  |  | 14 | 1. Yes |
|  |  |  |  |  |  |  | 2. No |
| **Suicidal attempt/Psychosis/Depression** | 9 (5.9) | 15 (9.8) | 10 (6.5) | 10 (6.5) | 109 (71.2) | 30 | 1. Suicidal Attempt |
|  |  |  |  |  |  |  | 2. Depression |
|  |  |  |  |  |  |  | 3. Depression and Suicidal Attempt |
|  |  |  |  |  |  |  | 4. Psychosis |
|  |  |  |  |  |  |  | 5. Nothing |
| **Magnitude of Administration** | 59 (33.5) | 102 (58) | 15 (8.5) |  |  | 7 | 1. Intense |
|  |  |  |  |  |  |  | 2. Moderate |
|  |  |  |  |  |  |  | 3. Mild |

Results expressed as number (percentage)

* = Severely intake of only Methamphetamine/Cannabis/Heroin along with or without other drugs

Others§ = Intake of other abusing substances (Alcohol/Fencidil/Sedatives) other than Methamphetamine/Cannabis/Heroin

Multiple† = Multiple drugs are taken almost equally at the same time.

polymorphism, no significant difference (p>0.05) was found in the age of onset among the three different genotypic groups (Table 5). On the other hand, the onset age was significantly different among the three genotypic groups (p<0.05) of DRD4 120 bp tandem duplication. Where 120 bp/120 bp carriers started using the drug at an earlier age compared to 120 bp/240 bp and 240 bp/240 bp carriers (Table 5). To compare pairwise differences among the age of onset of three different genotypic groups of DRD4 polymorphism, after post-hoc analysis the homozygous 120 bp/120 bp carriers have significantly (p<0.01) lower age of onset compared to 240 bp/240 bp carriers, and the mean difference was -4.431.

Further, all the addicted subjects were categorized into three groups- methamphetamine, heroin, and cannabis abusers to analyze whether these COMT and DRD4 polymorphisms can have any influence on the age of onset of individual substance abusers. However, no significant difference was found in the age of onset among three different genotypic groups of any specific substance abusers (S2 Table).

**Table 3. Genotype and allele distribution of COMT and DRD4 genes polymorphisms in substance abuser and control.**

| Gene | Genotype and Allele | n (%) | | OR (95% CI) | *p*-value |
|---|---|---|---|---|---|
| | | Substance Abuser (n = 183) | Control (n = 175) | | |
| COMT Val158Met | Val/Val | 62 (33.9) | 78 (44.6) | 1.0 (Ref.) | |
| | Val/Met | 79 (43.2) | 60 (34.3) | 1.66 (1.044–2.658) | <0.05 |
| | Met/Met | 42 (22.9) | 37 (21.1) | 1.43 (0.819–2.518) | ns |
| | Val (G) | 203 (55.5) | 216 (61.7) | 1.0 (Ref.) | |
| | Met (A) | 163 (44.5) | 134 (38.3) | 1.29 (0.959–1.752) | ns |
| DRD4 120bp VNTR | 120 bp/120 bp | 26 (14.2) | 11 (6.3) | 1.0 (Ref.) | |
| | 120 bp/240 bp | 71 (38.8) | 80 (45.7) | 0.37 (0.172–0.826) | <0.05 |
| | 240 bp/240 bp | 86 (47.0) | 84 (48.0) | 0.43 (0.193–0.937) | <0.05 |
| | 120 bp | 123 (33.6) | 102 (29.1) | 1.0 (Ref.) | |
| | 240 bp | 243 (66.4) | 248 (70.9) | 0.81 (0.593–1.110) | ns |

Chi-square tests were performed to calculate statistical significance. p< 0.05 was considered as a level of significance; OR: Odds ratio; CI: Confidence Interval; ns: not significant.

Again, because of the narrow age range in mean±SEM of the age of onset among different genotypic groups (according to Table 5), a breakdown of onset ages of abuse was done. Here the onset age range was divided into two categories: ≤20 and >20, and then association among different genotypic groups of COMT and DRD4 polymorphism was analyzed. However, no significant difference was found among different genotypic groups (S4 Table).

**Table 4. Association of genetic polymorphism of COMT and DRD4 genes with a risk of specific drug dependence.**

| Type of Dependence | Genotype | n (%) | | OR (95% CI) | *p*-value |
|---|---|---|---|---|---|
| | | Substance Abuser (n = 183) | Control (n = 175) | | |
| Methamphetamine (n = 78) | COMT | | | | |
| | Val/Val | 25 (32.1) | 78 (44.6) | 1.0 (Ref.) | |
| | Val/Met | 39 (50.0) | 60 (34.3) | 2.02 (1.109–3.670) | <0.05 |
| | Met/Met | 14 (17.9) | 37 (21.1) | 1.18 (0.569–2.449) | ns |
| Heroin (n = 45) | Val/Val | 16 (35.5) | 78 (44.6) | 1.0 (Ref.) | ns |
| | Val/Met | 21 (46.7) | 60 (34.3) | 1.71 (0.797–3.565) | |
| | Met/Met | 8 (17.8) | 37 (21.1) | 1.05 (0.419–2.742) | ns |
| Cannabis (n = 88) | Val/Val | 32 (36.4) | 78 (44.6) | 1.0 (Ref.) | ns |
| | Val/Met | 34 (38.6) | 60 (34.3) | 1.38 (0.768–3.502) | |
| | Met/Met | 22 (25.0) | 37 (21.1) | 1.45 (0.750–2.764) | ns |
| Methamphetamine (n = 78) | DRD4 | | | | |
| | 120bp/120bp | 7 (9.0) | 11 (6.3) | 1.0 (Ref.) | |
| | 120bp/240bp | 30 (38.5) | 80 (45.7) | 0.59 (0.222–1.546) | ns |
| | 240bp/240bp | 41 (52.5) | 84 (48.0) | 0.77 (0.268–1.994) | ns |
| Heroin (n = 45) | 120bp/120bp | 5 (11.1) | 11 (6.3) | 1.0 (Ref.) | ns |
| | 120bp/240bp | 18 (40.0) | 80 (45.7) | 0.50 (0.156–1.436) | |
| | 240bp/240bp | 22 (48.9) | 84 (48.0) | 0.58 (0.189–1.628) | ns |
| Cannabis (n = 88) | 120bp/120bp | 18 (20.4) | 11 (6.3) | 1.0 (Ref.) | <0.01 |
| | 120bp/240bp | 32 (36.4) | 80 (45.7) | 0.24 (0.107–0.558) | |
| | 240bp/240bp | 38 (43.2) | 84 (48.0) | 0.28 (0.124–0.616) | <0.01 |

Chi-square tests were performed to calculate statistical significance. p< 0.05 was considered as a level of significance; OR: Odds ratio; CI: Confidence Interval; ns: not significant.

**Table 5. Association of genetic polymorphisms of COMT and DRD4 genes with the age of onset of substance abuse.**

| Genotype | | n | Age of Onset (years) | *p*-value |
|---|---|---|---|---|
| **COMT Val158Met** | Val/Val | 62 | 18.9±0.9 | ns |
| | Val/Met | 74 | 20.2±0.8 | |
| | Met/Met | 42 | 18.5±0.9 | |
| **DRD4 120bp VNTR** | 120 bp/120 bp | 25 | 16.0±0.5 | <0.05 |
| | 120 bp/240 bp | 68 | 19.1±0.8 | |
| | 240 bp/240 bp | 85 | 20.5±0.8 | |

Results were expressed as mean±SEM; Mean age of onset for drug addiction was compared using Analysis of Variance (ANOVA); p<0.05 was considered as a level of significance; ns: not significant.

## Effect of COMT and DRD4 gene polymorphism on the pattern of drug habit

To analyze the effect of COMT and DRD4 polymorphism on the pattern of drug habit, the substance abused subjects were divided into two groups: subjects abused with 1–2 drugs and subjects abused with more than two (>2) drugs. According to Table 6, individuals carrying Val/Val variant are significantly more susceptible to more than two drugs compared to individuals carrying Val/Met variant (OR = 0.40; 95% CI = 0.206–0.804; p<0.05). But no significant difference was observed in genetic distribution between Val/Val and Met/Met variant containing groups. On the other hand, the genotypic distribution of DRD4 120bp polymorphism showed no significant influence on the pattern of drug habit.

## Relationship of the period of addiction with the type of drug used

To analyze the influence of methamphetamine, cannabis, and heroin individually on the period of addiction of substance abusers, they were categorized into two groups for each type of drug: Methamphetamine abusers and No-methamphetamine abusers; Cannabis abusers and No-Cannabis abusers; Heroin abusers and No-heroin abusers. As shown in Table 7, the period of addiction of no-methamphetamine abusers (10.2±0.7 years) was significantly (p<0.01) higher than the period of addiction of methamphetamine users (7.3±0.6 years). However, no significant influence of cannabis abuse was found in the period of addiction. Interestingly, the period of addiction to heroin abused group (11.62±1.01 years) was higher than the no-heroin abused group (8.01±0.5 years), which was statistically significant (p<0.01).

## Association of COMT and DRD4 polymorphism with a period of drug addiction of heroin abused subject

As the heroin abused patients were found to have a significantly higher period of addiction, the relationship in COMT and DRD4 gene polymorphism with the period of addiction of heroin abused subjects were analyzed. According to Table 8, the addiction period was significantly different among the three genotypic groups (p<0.001) of COMT Val158Met polymorphism. The Met/Met carriers had a longer addiction period to both Val/Val and Val/Met carriers. On the other hand, there was no significant variation in the period of addiction among the three different genotypic groups of DRD4 120 bp tandem duplication. However, no significant association of polymorphisms with the addiction period was found for both methamphetamine and cannabis abused subjects (S3 Table).

**Table 6. Association of genetic polymorphisms of COMT and DRD4 genes with the pattern of drug habit.**

| Gene | Genotype | Pattern of Drug Habit | | OR (95% CI) | *p*-value |
|---|---|---|---|---|---|
| | | 1–2 (n = 100) | >2 (n = 76) | | |
| COMT Val158Met | Val/Val | 29 (29.0) | 33 (43.4) | 1.0 (Ref.) | |
| | Val/Met | 50 (50.0) | 23 (30.3) | 0.40 (0.206–0.804) | <0.05 |
| | Met/Met | 21 (21.0) | 20 (26.3) | 0.83 (0.366–1.891) | ns |
| DRD4 120bp VNTR | 120 bp/120 bp | 16 (16.0) | 8 (10.5) | 1.0 (Ref.) | |
| | 120 bp/240 bp | 39 (39.0) | 29 (38.2) | 1.48 (0.584–4.009) | ns |
| | 240 bp/240 bp | 45 (45.0) | 39 (51.3) | 1.51 (0.597–4.026) | ns |

Chi-square tests were performed to calculate statistical significance. p< 0.05 was considered as a level of significance; OR: Odds ratio; CI: Confidence Interval; ns: not significant.

## Discussion

In Bangladesh, the effects of drug-related problems are gradually increasing from a social, economic, and medical perspective. Although research on drug abuse has been carried out globally, there is no information available on population genetics of COMT and DRD4 genetic variants for the substance abused in Bangladesh. Therefore, the present study is carried out to investigate the association of genetic polymorphism of COMT and DRD4 as a biomarker for increased risk of substance abuse in Bangladesh. Our result indicated these polymorphisms have a significant association with substance abuse and the age of onset, the pattern of drug habit, and period of addiction. According to the socio-demographic characteristics, a significantly higher percentage of addicted patients were less educated and unemployed compared to the control subjects (Table 1). Multiple logistic regression showed that unemployed, secondary educated, and low BMI containing subjects have a significantly higher tendency to be addicted than employed, graduate, and high BMI containing subjects. Therefore, according to our study, this lack of education, unemployment may have influence on the addictive status, and the studies of Hasam and Mushahid [31] and Islam et al. [32] also reported a higher percentage of unemployed abusers. Several studies suggest that peer influence and interest in a new experience are the main reasons behind addiction [33,34].

In contrast, our research suggests that maximum addicted subjects started taking drug due to peer influence (Table 2). Relapse behavior in substance abuse can be defined as returning to a state of addiction after a period of remission and treatment [35]. Although only 9.5% abusers showed relapse behavior, the rest of the abusers may also tend to show relapse behavior in the future (Table 2).

**Table 7. Relationship of type of drug used with period of addiction.**

| Type of Drug Abused | n | Period of Addiction (years) | p-value |
|---|---|---|---|
| Methamphetamine | 78 | 7.3±0.6 | <0.01 |
| No-Methamphetamine | 98 | 10.2±0.7 | |
| Cannabis | 88 | 9.4±0.6 | ns |
| No-Cannabis | 88 | 8.5±0.7 | |
| Heroin | 45 | 11.6±1.0 | <0.01 |
| No-Heroin | 131 | 8.0±0.5 | |

Results were expressed as mean±SEM; Unpaired t-test was performed and p<0.05 was considered as a level of significance; ns: not significant.

**Table 8. Association of the genetic polymorphisms of COMT and DRD4 genes with a period of drug addiction to heroin abused subjects.**

| Gene | Genotype | n | Period of Addiction (year) | p-value |
|---|---|---|---|---|
| **COMT Val158Met** | Val/Val | 16 | 9.6±1.4 | <0.001 |
| | Val/Met | 21 | 10.1±1.3 | |
| | Met/Met | 8 | 19.8±1.4 | |
| **DRD4 120bp VNTR** | 120 bp/120 bp | 5 | 15.2±2.5 | ns |
| | 120 bp/240 bp | 18 | 9.1±1.3 | |
| | 240 bp/240 bp | 22 | 12.9±1.6 | |

Results were expressed as mean±SEM; Mean for drug addiction was compared using Analysis of Variance (ANOVA); p<0.05 was considered as a level of significance; ns: not significant.

In this study, a significant relationship was found between the heterozygous genotype of COMT polymorphism and substance abuse (Table 3). Substance abuse subjects with COMT heterozygous mutant variants (Val/Met) showed a 1.66-fold increased risk of substance abuse compared to the control. According to the previous studies of COMT polymorphism and substance abuse in the different ethnic groups, the association of genetic variants of COMT Val158Met polymorphism was found in Turkish cannabis abusers [36], Taiwanese methamphetamine abusers [37], and Israeli Heroin abusers [38]. Therefore, the association of heterozygous mutant variants in our population suggests that this polymorphism may alter the ability of COMT enzyme to metabolize dopamine, and vulnerability to drugs is likely due to this altered enzyme activity. On the other hand, the homozygous mutant variant has no significant influence on the risk of substance abuse (Table 3), which is in agreement with the study of Oosterhuis et al. [39] as they found no association of the Met variant with Hispanic male heroin addicts.

Different studies previously reported genetic variants in the upstream DRD4 region with various neurobehavioral disorders such as novelty seeking, schizophrenia, attention-deficit/hyperactivity disorder, and substance abuse [21]. In our study, a significant relationship was found between both the derived genotype of DRD4 tandem duplication and risk for substance abuse where according to the odds ratio, both heterozygous (120bp/240bp) (OR = 0.37, 95% CI = 0.172–0.826) and homozygous (240 bp/240 bp) (OR = 0.43, 95% CI = 0.193–0.937) derived variants showed association with decreased risk of substance abuse (Table 3). It was also found that the longer allele (240bp/240bp) is found to be the most common in our population (both control and substance abused), which is consistent with other studies [4,40]. Therefore, our study suggests the longer allele (240bp/240bp) of the tandem duplication played a protective role in substance abuse, and according to a previous study by Kereszturi et al. the shorter allele (120bp/120bp) is found to be the risk allele in novelty seeking and other neuropsychiatric behavior [19]. These phenomena can be described by the fact that this tandem duplication in the upstream of initiation codon is the binding site of different transcriptional factors and this longer allele with a genetic tendency of lower transcriptional activity [41]. As a result, lower expression of DRD4 receptor may lead to decreased binding capacity of dopamine-to-dopamine receptor along with the additive effect of haplotype variations in the DRD4 promoter region may be related to the less risk of susceptibility to different neuropsychiatric disorders and substance abuse.

In Table 4, after categorizing the addicted subjects into three groups- methamphetamine, heroin, and cannabis abusers, we have found that genetic polymorphism may significantly affect a specific kind of drug abusers. In the case of COMT polymorphism, the methamphetamine abusers containing heterozygous Val/Met variant showed a significant association with

substance abuse compared to the control subjects. However, this polymorphism showed no association with heroin and cannabis abusers. As both high and low activity of COMT may have an adverse effect on reward processing, the association of COMT polymorphism with substance abuse is not likely a simple theory to describe [13]. It has been reported that the heterozygous Val/Met COMT encodes an enzyme with lower activity compared to the homozygous Val/Val genotype [9,13]. Low COMT enzyme activity is associated with high endogenous dopamine levels in the prefrontal cortex, leading to decreased dopaminergic neurotransmission in nucleus accumbens [42], and methamphetamine can induce dopamine release in nucleus accumbens [43]. This phenomenon is likely to be the reason behind the association of methamphetamine abuse with COMT polymorphism. Previously, no association was found between COMT Val158Met polymorphism and heroin or cannabis abuse from a meta-analysis study [13], but some studies found an association between this polymorphism and methamphetamine abuse [4,37].

In the DRD4 polymorphism case, only the cannabis abusers showed significant association with both heterozygous (OR = 0.24, 95% CI = 0.107–0.558) and homozygous (OR = 0.28, 95% CI = 0.124–0.616) derived variants where according to the odds ratio, the derived variants played a protective role. As cannabis may contribute to a decrease in the binding capacity of dopamine to dopamine receptors [44,45], the higher transcriptional activity of the shorter allele may lead to increased dopamine response by increasing dopamine receptors. Therefore, lower activity of longer allele (240bp/240bp) of this DRD4 polymorphism may contribute to the protective role in decreasing dopamine response of cannabis abusers.

Several studies reported the genotypic influence of substance abuse on the age of onset [23,46]. In our study, no significant difference was found on the age of onset for substance abuse among the three genotypic groups of COMT Val158Met polymorphism. A significant difference was found among the three different genotypic groups of DRD4 120 bp tandem duplication (Table 5). This study suggests the shorter allele (120 bp/120 bp) may be a risk factor for early-onset substance dependence. However, other substantial factors are associated with the onset age of substance abuse, such as family problems, socioeconomic class, depression, peer influence, and substance accessibility. Furthermore, many other SNPs across the DRD4 5' flanking region (promoter) may also effect the functions of DRD4, and even the onset ages of substance abuse [23].

In this study, we also found that the pattern of drug habit was associated with COMT Val158Met polymorphism (Table 6). Although the homozygous Met/Met variant showed no effect on the pattern of drug habit, individuals with homozygous Val/Val variant were significantly more prone to more than two drugs compared to individuals with heterozygous Val/Met variant (OR = 0.40; 95% CI = 0.206–0.804; p<0.05), which is inconsistent with the study of Vandenbergh et al. [26]. However, according to Table 6, no significant influence of the DRD4 heterozygote (120 bp/240 bp) allele and longer allele (240 bp/240 bp) wase observed on single or multi-drug taking behavior.

The duration of addiction is significant for the treatment of substance abused subjects. A longer addiction period was associated with an earlier onset age of addiction and relapse behavior and mental depression [47]. In this study, we have analyzed whether addiction to any specific drug is associated with a more extended period of addiction and found that heroin addicts had a significantly more extended period of addiction (Table 7). Opioid addiction can cause frequent relapse [48], and after opioid addiction, the brain functions abnormally without the presence of this drug [49]. This may be the reason behind the longer duration of addiction of heroin abusers.

Additional studies of the association of COMT Val158Met and DRD4 120bp VNTR polymorphism with heroin and other addictions were done to confirm and extend our findings. It

was found that the Met/Met variant containing heroin addicts had a longer addiction duration than Val/Val and Val/Met variants of COMT polymorphism (Table 8). Long term substance abuse results in reducing the number of dopamine receptors and decreased dopamine function and dopamine release in addicted subjects [50]. According to our study, the association of lower activity of COMT enzyme of Met/Met containing variants and consequently higher dopamine with more extended addiction is likely due to the decreased dopamine release and sensitivity after long-term addiction.

There are some limitations to this study. A self-reported questionnaire assessed drug-induced behavioral traits. The recall bias might have been undoubtedly involved when the data was collected based on the recall of subjective responses that occurred in a distant past. Again, our study population had a relatively small sample size from one center and consisted of only the Bangladeshi male race. In summary, while this study has some restrictions, it is the first research in which the relationship between COMT Val158Met and DRD4 120bp VNTR variant and substance abuse in the Bangladeshi population were analyzed. Although the results were significant, this was a preliminary evaluation, and studies in larger samples are needed to confirm these promising results.

In future research more association studies with more candidate genes involving a larger sample size and more variables, such as gene-gene interactions and interaction between gene and environment, are required to generate a comprehensive understanding of the actual genetic association and underlying mechanism of substance abuse. Further research is also required to prospectively evaluate the role of genetic variants in the development of clinical manifestations of substance abuse, which may contribute to the rational therapeutic treatment.

In conclusion, the present study has demonstrated a significant association of COMT heterozygous variant polymorphism (Val/Met) as well as both homozygous and heterozygous variants of DRD4 120 bp tandem duplication with risk of substance abuse among the Bangladeshi population.

## Supporting information

**S1 Table. Association of COMT polymorphism in dominant and recessive model.**
(DOCX)

**S2 Table. Association of genetic polymorphisms of COMT and DRD4 genes with the age of onset for specific substance abuse.**
(DOCX)

**S3 Table. Association of the genetic polymorphisms of COMT and DRD4 genes with a period of drug addiction to Methamphetamine and Cannabis abused subjects.**
(DOCX)

**S4 Table. Association of genetic polymorphisms of COMT and DRD4 genes with the age of onset of substance abuse (at different age range).**
(DOCX)

## Acknowledgments

Authors thank physicians and nurses of Central Drug Addiction Treatment Center, Dhaka and Dhaka Medical College Hospital and Bangabandhu Sheikh Mujib Medical University, Dhaka, Bangladesh for counseling of study subjects and their technical assistance during blood collection. We also thank all the study subjects for participating in this study. Authors are especially thankful to Dr. Syed Emamul Hossain, Chief Consultant, Central Drug Addiction

Treatment Center, Department of Narcotics Control, Ministry of Home Affairs, Government of the People's Republic of Bangladesh, for allowing and helping to collect blood samples from the drug-addicted patients.

## Author Contributions

**Conceptualization:** Jahanara Akter Sonia, Yearul Kabir.

**Data curation:** Jahanara Akter Sonia.

**Formal analysis:** Jahanara Akter Sonia, Tohfa Kabir.

**Investigation:** Jahanara Akter Sonia, Tohfa Kabir.

**Project administration:** Tohfa Kabir, M. M. Towhidul Islam, Yearul Kabir.

**Resources:** Yearul Kabir.

**Supervision:** M. M. Towhidul Islam, Yearul Kabir.

**Writing – original draft:** Jahanara Akter Sonia.

**Writing – review & editing:** M. M. Towhidul Islam, Yearul Kabir.

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
