## [Decision Letter · Decision Letter 0]

13 Aug 2020

PONE-D-20-16794

Catechol-O-methyltransferase and dopamine receptor D4 gene variants: possible association with substance abuse in Bangladeshi male

PLOS ONE

Dear Dr. Kabir,

Thank you for submitting your manuscript to PLOS ONE. After careful consideration, we feel that it has merit but does not fully meet PLOS ONE’s publication criteria as it currently stands. Therefore, we invite you to submit a revised version of the manuscript that addresses the points raised during the review process.

Please address reviewers' comments thoroughly. Your revision will go back to the same reviewers for satisfaction.

We look forward to receiving your revised manuscript.

Kind regards,

Zhicheng Lin, Ph.D.

Academic Editor

PLOS ONE

Journal Requirements:

2. Please ensure that your entire ethics statement shown in the submission form 'Institutional Ethical Review Committee of Department of Biochemistry and Molecular biology, University of Dhaka approved the study (No. BMBDU-ERC/EC/18/016). Written consent obtained from all the study participants' is included in the manuscript. In addition, please clarify who provided consent in the case of minors.

Reviewers' comments:

Reviewer's Responses to Questions

**Comments to the Author**

1. Is the manuscript technically sound, and do the data support the conclusions?

Reviewer #1: No

Reviewer #2: Yes

Reviewer #3: Partly

2. Has the statistical analysis been performed appropriately and rigorously? 

Reviewer #1: No

Reviewer #2: Yes

Reviewer #3: Yes

3. Have the authors made all data underlying the findings in their manuscript fully available?

Reviewer #1: Yes

Reviewer #2: Yes

Reviewer #3: Yes

4. Is the manuscript presented in an intelligible fashion and written in standard English?

Reviewer #1: Yes

Reviewer #2: Yes

Reviewer #3: No

5. Review Comments to the Author

Reviewer #1: Dear Authors,

Almost all odds ratio (OR) presentations are wrong. Just Table 6 is correct as the subjects abused with 1-2 drugs is the reference. But, in all other tables, the given ORs are not the ones which should be calculated according to reference category, actually they are the ORs of the (given) reference category if the corresponding genotype is considered as reference. If, ORs and CIs get reversed as (1/OR and 1/Upper Limit-1/Lower Limit), they would be correct. So, the ORs and confidence intervals must be corrected for Table 3 and 4. Also, due to these major mistakes all the inferences are also wrong. For example, on the lines 321-323 “the heterozygote genotype plays a protective role according to the odds ratio (0.60) and that drug addicts have a higher frequency of COMT Val/Val variant” inference is totally wrong. Because the OR of Val/Met heterozygotes is 1.656 (%95CI: 1.302-2.659) when Val/Val homozygotes are considered as reference, and drug addicts have a lower frequency of Val/Val variant. Since, the manuscript is based on reverse OR estimations, the results and discussion should be totally adjusted. Also, under the tables it is stated that Fisher’s exact test was used for determining the significance level of ORs, logistic regression analysis already gives the individual p-values for each genotype (or category) so no need to perform another test procedure.

Also, in discussion it is stated that “lack of education and unemployment may have a high influence on the addictive status” (lines 311-312), so to estimate the risk among the genotypes (or groups), it is better to use multivariate logistic regression analysis instead of univariate analysis.

Under statistical analysis section, it is stated that quantitative data were expressed as mean±SEM, but under Table 1 it is written as SD. Actually, they all should be standard deviation (SD) instead of standard error of mean (SEM). Also, the authors used parametric test procedures like independent samples t-test or one-way analysis of variance, but they did not give any information about the normality assumption or homogeneity of variances. If data met these assumptions, after one-way ANOVA a post-hoc analysis need to be performed to determine pairwise differences.

Reviewer #2: Comment on Manuscript PONE-D-20-16794: “Catechol-O-methyltransferase and dopamine receptor D4 gene variants: possible association with substance abuse in Bangladeshi male”

With this paper, the authors sought to investigate the association of genetic polymorphism of COMT and DRD4 as a biomarker for increased risk of substance abuse in Bangladesh. To this end used 183 male substance-dependent patients belonging to the age group of 15-49 years and 175 control subjects belonging to the same age group. The authors showed an association between the investigated polymorphisms (COMT and DRD4) with substance abuse, as well as the age of onset, the pattern of drug habit (how many types of substance), and the period of addiction. Also, the study provides interesting information about the relationship between substance use and sociodemographic characteristics. For example, according to the sociodemographic characteristics, a significantly higher percentage of addicted patients were illiterate and unemployed compared to the control subject. These results show that genetic variability may influence the susceptibility to risk of substance abuse as well as addictive characteristics. In an overview, the study provides interesting information on the relationship between substance use and genetic polymorphisms (biological factor), as well as with sociodemographic characteristics for the investigated population. The article too is very well written. However, some information related to ethical care needs to be informed, information about the statistical method needs to be more detailed and the study's findings need to be better discussed for a better understanding of the reader. A couple of issues I would like the authors to give a second thought:

Abstract:

Lines 38-40 – Does the information contained in these lines agree with what is presented in the results and the discussion?

Introduction: The logic was very well conducted. The authors were able to expose the research problem.

I suggest dividing the first paragraph into two parts.

Line 53 - has more space between factors and [4].

Line 58 - there is an additional parenthesis next to the reference [6].

Line 80 - is missing a space between the receiver and (DRD4).

Line 81 – Put a comma after the quotes.

Line 97 - Put "the" before age.

Lines 62-63 - Is it necessary to explain the mechanism involved in this process or to make it clearer?

Methods: It is well detailed about the sample and the genotyping processes, but the statistical procedures still need to be better detailed.

Why did the authors choose to investigate only male individuals?

Was the option due to the greater frequency of substance abuse being male? If yes, provide epidemiological data on this frequency by sex in the introduction. Or was it because of the effect of sex on genotype modulations? Make it clear.

Did participants under the age of 18 sign the free and informed consent form?

Can the quality of the figures be improved? (Figures 1a and 1b)

I consider it extremely important that the authors make the statistical tests clearer (for example, show how the logistic regression was performed) and what the values in the tables represent, mainly about Odds ratios.

Results: They are well organized and detailed, but they can be better details or the values in the tables represent, mainly about Odds ratios.

Lines 185-187 - it is written “Here, the frequency of the Val/Val genotype is significantly higher in addicted subjects than control subjects (OR=0.60, 95% CI=0.376-0.957).”, but wouldn't it be the Val/Met genotype? According to the data in table 3. This also does not match the writing of the abstract.

Place table 4 on line 201 and remove from lines 207 and 209.

Table 4 shows the association of the genetic polymorphism of the COMT and DRD4 genes with the risk of specific drug dependence. What was the intention of the authors when presenting the data of the control group? The intention was to categorize dependent individuals into three groups (i.e., users of methamphetamine, heroin, and cannabis) and relate them to genetic polymorphisms. And so, identifying whether there would be a significant effect on a specific type of drug user.

Tables – Standardize mainly on tables 5 and 8.

Discussion: I think the authors could have discussed the findings better.

Line 333 – Nucleus accumbens? If yes, I suggest rephrasing the sentence to “mesolimbic dopaminergic centers and dopamine release to the nucleus accumbens”

The authors brought good arguments for the relationship between the COMT polymorphism and the risk of substance abuse, at the end of the second paragraph. However, the arguments for the relationship between DRD4 tandem repeat polymorphism and the risk of substance abuse have been little explored.

What could be the possible explanations for the relationship between the COMT polymorphism and the use of methamphetamine? More specifically, what would be a possible explanation for this finding “the methamphetamine abusers containing heterozygous Val/Met variant showed a significant association with substance abuse compared to Val/Val containing variants”?

Try to provide explanations also for the findings regarding DRD4 tandem repeat polymorphism and cannabis use.

The authors point out some limitations of the study, including the sample being composed only of male individuals. As I said before, I would like to understand why the authors choose only male individuals.

Lines 386-387 – What do the authors mean by developing more rational therapeutic approaches?

In the penultimate paragraph, the authors seek to explain the long duration of addiction in heroin users. However, they do not explain this relationship with the genetic variants of the COMT polymorphism (i.e., Met/Met genotype). Why would a genotype that is associated with a lower activity of the COMT enzyme and consequently a higher amount of dopamine in the synaptic cleft is associated with a longer duration of addiction in heroin users?

The authors could suggest possible future research.

Reviewer #3: A thorough and critical review of the grammar and writing should be undertaken.

It would be useful to know the population genetics of these genes/variants? In Bangladesh? In a healthy population?

What about comorbid conditions, specifically mental health disorders? Pain conditions?

Line 30

Only het variant COMT associated with risk of substance abuse

Not homozygous? No association with either allele? This is strange and requires an explanation.

Did you look at COMT haplotypes? It has been demonstrated that COMT haplotypes and not the Val158Met locus alone may be important in addiction, as well as pain sensitivity.

Lines 41-43

Do the genetics say anything about period of addiction? Or is that just the substance? It has been suggested that heroin is simply more addictive than many other substances.

Lines 93-94: “high freq of homozygous valine variant of COMT polymorphism was found in polysubstance abuse.” Authors report only het form of COMT assoc with abuse

Line 226: earliest age of onset = 10 yo. Line 228 states “early age of adulthood (9.3). That’s younger than the earliest described age of onset.

Fig 1A lanes are mislabeled. Move arrows so they’re not blocking lanes for 16 and 17 (current numbering).

Fig 1B move arrows so they’re not in sample lanes

Add 100 bp ladder source to methods

Table 3: het COMT assoc with substance abuse. Why isn’t one of the alleles associated? This conclusion requires an explanation.

Table 5: Seems like narrow age ranges, esp considering the population range of the sample population. May consider age ranges when looking for significance or provide a breakdown of the ages in the target population. Maybe state something about how age itself is a risk factor for substance abuse. In fact, it is an integral part of the ORT.

Discussion: need better mechanism of explanation.

What about treatment? Treatment failure?

6. PLOS authors have the option to publish the peer review history of their article (what does this mean?). If published, this will include your full peer review and any attached files.

Reviewer #1: No

Reviewer #2: No

Reviewer #3: No

---

## [Author Response · Author response to Decision Letter 0]

21 Nov 2020

A point-by-point reply to Reviewers comments:    

Reviewer # 1

Comment: Almost all odds ratio (OR) presentations are wrong. Just Table 6 is correct as the subjects abused with 1-2 drugs is the reference. But, in all other tables, the given ORs are not the ones which should be calculated according to reference category, actually they are the ORs of the (given) reference category if the corresponding genotype is considered as reference. If, ORs and CIs get reversed as (1/OR and 1/Upper Limit-1/Lower Limit), they would be correct. So, the ORs and confidence intervals must be corrected for Table 3 and 4. Also, due to these major mistakes all the inferences are also wrong. For example, on the lines 321-323 “the heterozygote genotype plays a protective role according to the odds ratio (0.60) and that drug addicts have a higher frequency of COMT Val/Val variant” inference is totally wrong. Because the OR of Val/Met heterozygotes is 1.656 (%95 CI: 1.302-2.659) when Val/Val homozygotes are considered as reference, and drug addicts have a lower frequency of Val/Val variant. Since, the manuscript is based on reverse OR estimations, the results and discussion should be totally adjusted. 

Author’s Reply: Yes, we agree with the reviewer that the odds ratio presented was incorrect because data were entering a different arrangement during data analysis. Now we corrected the ORs and confidence intervals for Tables 3 and 4 as well as the results and discussion according to the new inferences in the manuscript. The corrected results were included in Lines: 207-217 & 232-233 and in Tables 3 and 4. Further, the discussion part was also adjusted according to new inferences (Lines: 352-358 and 362-378).

Comment: Also, under the tables, it is stated that Fisher’s exact test was used for determining the significance level of ORs; logistic regression analysis already gives the individual p-values for each genotype (or category), so no need to perform another test procedure.

Author’s Reply: We agree with the reviewer that no need to perform two tests for genotype (or category) analysis. We did not perform a logistic regression analysis, which was previously mentioned in the manuscript. Only Fisher’s exact test was performed for genotype analysis, corrected in the revised manuscript (Lines: 153-155). 

Comment: Also, in the discussion it is stated that “lack of education and unemployment may have a high influence on the addictive status” (lines 311-312), so to estimate the risk among the genotypes (or groups), it is better to use multivariate logistic regression analysis instead of univariate analysis.

Author’s Reply: According to the reviewer's suggestion, we have performed multivariate logistic analysis where genotype was taken as a primary variable and other demographic variable as co-variate. Due to the unavailability of illiterate subjects in the control group and to perform multivariate analysis, we combined drug addicted illiterate subjects with secondary educated subjects and compared with the graduate subjects. The outcome of this result is presented in Lines: 167-169. The addition result of BMI due to multivariate logistic analysis was also shown in Lines: 171-173. This result is also discussed in Lines: 338-340.

Comment: Under the statistical analysis section, it is stated that quantitative data were expressed as mean±SEM, but under Table 1 it is written as SD. Actually, they all should be standard deviation (SD) instead of the standard error of the mean (SEM). 

Author’s Reply: Thank you for indicating a serious mistake. SD in Table 1 is a typing mistake. We cross-checked all data; mean±SEM was presented in all cases, including Table 1. Accordingly, the footnote of Table 1 was corrected. 

Comment: Also, the authors used parametric test procedures like independent samples t-test or one-way analysis of variance, but they did not give any information about the normality assumption or homogeneity of variances. If data met these assumptions, after one-way ANOVA a post-hoc analysis need to be performed to determine pairwise differences. 

Author’s Reply: In this study, after one-way ANOVA, a posthoc analysis was done (as mentioned by the reviewer) to compare pairwise analysis (Lines: 159-160), and the result was given (Lines: 259-263 for Table 5 and Lines: 321-322 for Table 8) as our data meet the normality assumption (now mentioned in the revised manuscript, Line: 157) according to below reports. 

As reported [1], according to the central limit theorem, (a) if the sample data are approximately normal, then the sampling distribution too will be normal; (b) in large samples (> 30 or 40), the sampling distribution tends to be normal, regardless of the shape of the data [2, 3]; and (c) means of random samples from any distribution will themselves have normal distribution [4].

Besides, with large enough sample sizes (> 30 or 40), the violation of the normality assumption should not cause major problems [5]; this implies that we can use parametric procedures even when the data are not normally distributed [3].

 1. Ghasemi A, Zahediasl S. Normality Tests for Statistical Analysis: A Guide for Non-Statisticians. Int J Endocrinol Metab. 2012;10(2):486-489. DOI: 10.5812/ijem.3505

 2. Field A. Discovering statistics using SPSS. 3rd ed. London: SAGE publications Ltd; 2009. p. 822. 

 3. Elliott AC, Woodward WA. Statistical analysis quick reference guidebook with SPSS examples. 1st ed. London: Sage Publications; 2007.

 4. Altman DG, Bland JM. Statistics notes: the normal distribution. BMJ 1995;310(6975):298.

 5. Pallant J. SPSS survival manual, a step by step guide to data analysis using SPSS for windows. 3rd ed. Sydney: McGraw Hill; 2007. p. 179-200

Reviewer # 2 

Comment: Comment on Manuscript PONE-D-20-16794: “Catechol-O-methyltrans-ferase and dopamine receptor D4 gene variants: possible association with substance abuse in Bangladeshi male.”

With this paper, the authors sought to investigate the association of genetic polymorphism of COMT and DRD4 as a biomarker for increased risk of substance abuse in Bangladesh. To this end used 183 male substance-dependent patients belonging to the age group of 15-49 years and 175 control subjects belonging to the same age group. The authors showed an association between the investigated polymorphisms (COMT and DRD4) with substance abuse, as well as the age of onset, the pattern of drug habit (how many types of substance), and the period of addiction. Also, the study provides interesting information about the relationship between substance use and sociodemographic characteristics. For example, according to the sociodemographic characteristics, a significantly higher percentage of addicted patients were illiterate and unemployed compared to the control subject. These results show that genetic variability may influence the susceptibility to risk of substance abuse as well as addictive characteristics. In an overview, the study provides interesting information on the relationship between substance use and genetic polymorphisms (biological factor), as well as with sociodemographic characteristics for the investigated population. The article too is very well written. However, some information related to ethical care needs to be informed, information about the statistical method needs to be more detailed and the study's findings need to be better discussed for a better understanding of the reader. A couple of issues I would like the authors to give a second thought:

Abstract:

Comment: Lines 38-40 – Does the information contained in these lines agree with what is presented in the results and the discussion?

Author’s Reply: Yes, lines 38-40 (line 39-41 in the revised manuscript) agree with the results and discussion section's findings. To make it clearer, we replaced those lines with “the frequency of Val/Val genotype is higher in polysubstance abused (>2 drugs) subjects (p<0.05) compared to heterozygous Val/Met containing variants” (Lines: 39-41).

Introduction: The logic was very well conducted. The authors were able to expose the research problem.

Comment: I suggest dividing the first paragraph into two parts.

Line 53 - has more space between factors and [4].

Line 58 - there is an additional parenthesis next to the reference [6].

Line 80 - is missing a space between the receiver and (DRD4).

Line 81 – Put a comma after the quotes.

Line 97 - Put "the" before age.

Author’s Reply: According to the suggestion of the reviewer, the first paragraph was divided into two, and corrections were done at Lines 53 (line 55 in the revised manuscript), 58 (line 61 in the revised manuscript), 80 (line 81 in the revised manuscript), 81 (line 82 in the revised manuscript) and 97.

Comment: Lines 62-63 - Is it necessary to explain the mechanism involved in this process or to make it clearer?

Author’s Reply: Lines 62-63, is not necessary to explain the mechanism involved in this process, so we removed this line.

Comment: Methods: It is well detailed about the sample and the genotyping processes, but the statistical procedures still need to be better detailed.

Author’s Reply: In the revised manuscript, we explained all statistical procedures in detail in the Statistical analysis part of the methods and materials (Lines: 151-161). 

Comment: Why did the authors choose to investigate only male individuals?

Limitation- Lines 409-410

Was the option due to the greater frequency of substance abuse being male? If yes, provide epidemiological data on this frequency by sex in the introduction. Or was it because of the effect of sex on genotype modulations? Make it clear.

Author’s Reply: According to the reviewer's suggestion, the reasoning behind investigating only male substance abused subjects is now included in the introduction (Lines: 53-54) and in the method section (lines: 111-113). 

Comment: Did participants under the age of 18 sign the free and informed consent form

Author’s Reply: For under-age 18 yrs, their parent or legal guardian signs the written consent. Now, this information was incorporated in the method section under the ethical permission statement (Lines: 117-118).

Comment: Can the quality of the figures be improved? (Figures 1a and 1b).

Author’s Reply: Improved quality Figures and better resolution of figures are now provided in the revised manuscript. 

Comment: I consider it extremely important that the authors make the statistical tests clearer (for example, show how the logistic regression was performed) and what the values in the tables represent, mainly about Odds ratios.

Author’s Reply: In the revised manuscript, we explained all the statistical procedures in detail and made the statistical tests clearer (Lines: 153-155). The odd ratio was not calculated by using logistic regression analysis, as mentioned in the previous manuscript. Only Fisher’s exact test was performed for genotype analysis (odds ratio), which was corrected in the revised manuscript. The odds ratio value indicates the higher/lower risk of substance abuse with any particular genotype compared to the reference group.

Results: They are well organized and detailed, but they can be better details or the values in the tables represent, mainly about Odds ratios.

Comment: Lines 185-187 - it is written “Here, the frequency of the Val/Val genotype is significantly higher in addicted subjects than control subjects (OR=0.60, 95% CI=0.376-0.957).”, but wouldn't it be the Val/Met genotype? According to the data in table 3. This also does not match the writing of the abstract.

Author’s Reply: Yes, we agree with the reviewer; it would be Val/Met genotype. According to newly calculated odds ratio and confidence intervals, the frequency of the Val/Met genotype (not Val/Val genotype) is significantly higher in addicted subjects than in control subject, which was now corrected in the revised manuscript (Lines: 210-211 and in Table 3). The abstract was also modified accordingly (Lines: 30-32).

Comment: Place table 4 on line 201 and remove from lines 207 and 209.

Author’s Reply: The word Table 4 was placed and removed according to the reviewer's suggestion.

Comment: Table 4 shows the association of the genetic polymorphism of the COMT and DRD4 genes with the risk of specific drug dependence. What was the intention of the authors when presenting the data of the control group? The intention was categorized dependent individuals into three groups (i.e., users of methamphetamine, heroin, and cannabis) and relate them to genetic polymorphisms. And so, identifying whether there would be a significant effect on a specific type of drug user.

Tables – Standardize mainly on tables 5 and 8.

Author’s Reply: Yes, the intention was to categorize drug-dependent subjects into three groups and analyze if any specific dependent group has any association with individual drugs compared to the control group (Table 4).

According to the suggestion of the reviewer, Tables 5 and 8 were standardized. Table 5 was standardized according to (in line with or similar to) Table 4, where the drug-dependent subjects were categorized into three groups to analyze the association with age of onset of substance abuse. However, no significant difference was found for any specific type of drug, and the result is given in Line 264-268 and the S2 Table.

In the previous manuscript, Table 8 investigated the association of period of addiction, specifically with heroin abused subjects. To standardize table 8 according to table 4, in the revised manuscript, a similar analysis was done for methamphetamine and cannabis abused groups, but no significant association of polymorphisms with the addiction period was found (result discussed in lines 319-322 and in S3 Table).

Discussion: I think the authors could have discussed the findings better.

Author’s Reply: Now, the findings were discussed in a better way according to the results obtained by analyzing the data in a more appropriate way in the revised manuscript. 

Comment: Line 333 – Nucleus accumbens? If yes, I suggest rephrasing the sentence to “mesolimbic dopaminergic centers and dopamine release to the nucleus accumbens”

Author’s Reply: In the revised manuscript, the point was removed because of newly generated data due to the different statistical analysis approaches. We rewrite the discussion with an appropriate new interpretation of the study findings (Lines 352-358). 

Comment: The authors brought good arguments for the relationship between the COMT polymorphism and the risk of substance abuse, at the end of the second paragraph. However, the arguments for the relationship between DRD4 tandem repeat polymorphism and the risk of substance abuse have been little explored.

Author’s Reply: In the revised manuscript, the relationship between DRD4 tandem repeat polymorphism and the risk of substance abuse has been explored more based on previous studies of this polymorphism's association with neuro-psychiatric behavior and substance abuse (lines: 362-378). 

Comment: What could be the possible explanations for the relationship between the COMT polymorphism and the use of methamphetamine? More specifically, what would be a possible explanation for this finding “the methamphetamine abusers containing heterozygous Val/Met variant showed a significant association with substance abuse compared to Val/Val containing variants”?

Author’s Reply: As suggested, a more detailed explanation is given for the association of methamphetamine abuse and COMT polymorphism in the revised manuscript (Lines: 384-394). 

Comment: Try to provide explanations also for the findings regarding DRD4 tandem repeat polymorphism and cannabis use.

Author’s Reply: As suggested, we provided possible explanations for the findings regarding DRD4 tandem repeat polymorphism and cannabis use with corrected odds ratio and confidence intervals in lines 395-402.

Comment: The authors point out some limitations of the study, including the sample being composed only of male individuals. As I said before, I would like to understand why the authors choose only male individuals.

Author’s Reply: The reason behind the sample being composed of only male subjects were given in a previous reply and mentioned in the introduction (Lines: 53-54) and method sections (Lines: 111-113) in the revised manuscript.

Comment: Lines 386-387 – What do the authors mean by developing more rational therapeutic approaches? 

Author’s Reply: In the revised manuscript we have replaced the line stating more rational therapeutic approaches with a new para stating possible future research to evaluate the role of genetic variants in the development of clinical implications (Lines: 444-449). 

Comment: In the penultimate paragraph, the authors seek to explain the long duration of addiction in heroin users. However, they do not explain this relationship with the genetic variants of the COMT polymorphism (i.e., Met/Met genotype). Why would a genotype that is associated with a lower activity of the COMT enzyme and consequently a higher amount of dopamine in the synaptic cleft is associated with a longer duration of addiction in heroin users? 

Author’s Reply: In the revised manuscript, we explained the reason behind the Met/Met genotype relationship with a longer addiction period (Line 429-434). Long term substance abuse results in reducing the number of dopamine receptors as well as decreased dopamine function and dopamine release in addicted subjects (ref. 50 in MS). Lower activity of COMT enzyme of Met/Met containing variants and consequently higher amount of dopamine with longer addiction period is likely due to the decreased dopamine release and sensitivity after long term addiction.

Comment: The authors could suggest possible future research.

Author’s Reply: The possible future research is included within Line 444-449. 

Reviewer # 3: 

Comment: A thorough and critical review of the grammar and writing should be undertaken.

Author’s Reply: As suggested by the reviewer, a thorough and critical review of the grammar and writing was undertaken.

Comment: It would be useful to know the population genetics of these genes/variants? In Bangladesh? In a healthy population? 

Author’s Reply: There was no information on population genetics of COMT and DRD4 genetic variants for the Bangladeshi population (both for addicted and healthy population) (Lines: 330-332). Therefore, we discussed other countries' population genetics data in the discussion section (Lines: 352-358 and Lines: 362-378). To our knowledge, this study is the first to make a report on the association of these polymorphisms with substance abuse in Bangladesh. 

Comment: What about comorbid conditions, specifically mental health disorders? Pain conditions? 

Author’s Reply: During sample collection, we have collected information on mental health disorders- depression, suicidal attempt, and psychosis, and this information is mentioned in Table 2 of the manuscript. Data on pain conditions was not obtained (not included in the questionnaire), as our study focused on the genetic polymorphisms of genes with addiction.

Comment: Line 30: Only het variant COMT associated with risk of substance abuse

Not homozygous? No association with either allele? This is strange and requires an explanation.

Did you look at COMT haplotypes? It has been demonstrated that COMT haplotypes and not the Val158Met locus alone may be important in addiction, as well as pain sensitivity. 

Author’s Reply: This is an important question and also raised our concerns. Since we did not find any association with homozygous Met/Met variant or Methionine allele alone, we stratified our analysis based on the different inheritance models. Further, we analyzed the association of COMT polymorphism in dominant and recessive models. It was found that in the dominant model, the combined GA+AA genotype appeared to have an increased susceptibility towards substance abuse, while the effect was not statistically significant in the recessive model (S1 Table). This, in part, could explain why we did not observe any association at the allelic level. However, this cannot be the only cause for missing association in the homozygous genotypes, but the number of people in the AA homozygous are a bit smaller and could explain the lack of statistical powers to ascertain any significant association.

Unfortunately, we did not look at COMT haplotype. We agree with the reviewer that there is a possibility that COMT haplotypes and not the Val158Met locus alone may be responsible for addiction in the Bangladeshi population as reported by Jugurnauth et al. (2011) (ref. 37 in MS) and Lohoff et al. (2008) (ref. 15 in MS) in different ethical populations. However, since we genotyped only this Val158Met polymorphism, we could not build our population's haplotype. This could be a direction for our future studies to genotype all the possible functionally significant tagged SNPs in the COMT genes to make our population's haplotype.

Comment: Lines 41-43. Do genetics say anything about the period of addiction? Or is that just the substance? It has been suggested that heroin is simply more addictive than many other substances.

Author’s Reply: Both the substance and genetics have an effect on the period of addiction. As in the manuscript, lines 41-43 mentioned the effect of substance abuse on the period of addiction by stating Heroin abusers were having a longer period of addiction; in the revised manuscript, the effect of genetics in the period of addiction is also discussed (Lines: 43-45). 

Comment: Lines 93-94: “high freq of homozygous valine variant of COMT polymorphism was found in polysubstance abuse.” Authors report only het form of COMT assoc with abuse

Author’s Reply: Lines 93-94: we just reported the findings of Vandenbergh et al., 1997 (Ref. 26 in MS) that “high freq of homozygous valine variant of COMT polymorphism was found in polysubstance abuse." Similarly, we also found the frequency of homozygous Val/Val genotype (not the heterozygous form) is higher in polysubstance abused (>2 drugs) subjects compared to Val/Met variant, which is mentioned in the result section (Lines: 285-287 and in Table 6).

Comment: Line 226: earliest age of onset = 10 yo. Line 228 states “early age of adulthood (9.3). That’s younger than the earliest described age of onset.

Author’s Reply: In line 228 of the previous manuscript, it was a typo error, which is now corrected as “early age of adulthood (19.3±0.5 years, mean±SEM age of onset of all abuser) (Line 254 in the revised manuscript). In line 253 (line 226 in the previous manuscript), the earliest age of onset indicated the lowest age of onset in the study.

Comment: Fig 1A lanes are mislabeled. Move arrows so they’re not blocking lanes for 16 and 17 (current numbering).

Fig 1B move arrows so they’re not in sample lanes

Add 100 bp ladder source to methods.

Author’s Reply: According to the reviewer's suggestion, the labeling patterns of the figures (1A and 1B) are changed. Also, the ladder source was included (Line: 144).

Comment: Table 3: het COMT assoc with substance abuse. Why isn’t one of the alleles associated? This conclusion requires an explanation.

Author’s Reply: As replied previously, we did not find any association with homozygous Met/Met variant or Methionine allele alone; we wanted to stratify our analysis based on the different models of inheritance. Further, we analyzed the association of COMT polymorphism in dominant and recessive models. It was found that in the dominant model, the combined GA+AA genotype appeared to have an increased susceptibility towards substance abuse, while the effect was not statistically significant in the recessive model (S1 Table). This, in part, could explain why we did not observe any association at the allelic level. However, this cannot be the only cause for missing association in the homozygous genotypes, but the number of people in the AA homozygous are a bit smaller and could explain the lack of statistical powers to ascertain any significant association.

Comment: Table 5: Seems like narrow age ranges, esp considering the population range of the sample population. May consider age ranges when looking for significance or provide a breakdown of the ages in the target population. Maybe state something about how age itself is a risk factor for substance abuse. It is an integral part of the ORT.

Author’s Reply: Possibly, age itself is not a risk factor for drug addiction as drug addiction was found in a large range of age (10-44 years). Moreover, we did not find any previous report indicating age as a risk factor for drug addiction. From our findings, we assume that the onset age of substance abuse can act as a risk factor (according to Table 5).

In our sample population, the age of onset range was from 10 to 44 years. Considering the age range of our study population and looking for the significance, we divided onset age of substance abusers into ≤20 years and >20 years, assuming 20 years old as adulthood age, but no significant difference was found among different genotypic groups of COMT and DRD4 polymorphisms (S4 Table and result explained in Lines 275-279 in the manuscript)

Comment: Discussion: need better mechanism of explanation.

Author’s Reply: In the revised manuscript, better interpretation and explanation of generated data were provided with the possible mechanism.

Comment: What about treatment? Treatment failure?

Author’s Reply: The treatment and their failure in relation to polymorphism was not the aims/focus of this study. In the future, we will study the treatment regime and outcome in relation to genetic polymorphism.

---

## [Decision Letter · Decision Letter 1]

10 Dec 2020

PONE-D-20-16794R1

Catechol-O-methyltransferase and dopamine receptor D4 gene variants: possible association with substance abuse in Bangladeshi male

PLOS ONE

Dear Dr. Kabir,

Thank you for submitting your manuscript to PLOS ONE. After careful consideration, we feel that it has merit but does not fully meet PLOS ONE’s publication criteria as it currently stands. Therefore, we invite you to submit a revised version of the manuscript that addresses the points raised during the review process.

Reviewers have acknowledged the improvement but few concerns still remain. Please address carefully and make sure of satisfying results.

We look forward to receiving your revised manuscript.

Kind regards,

Zhicheng Lin, Ph.D.

Academic Editor

PLOS ONE

Reviewers' comments:

Reviewer's Responses to Questions

**Comments to the Author**

1. If the authors have adequately addressed your comments raised in a previous round of review and you feel that this manuscript is now acceptable for publication, you may indicate that here to bypass the “Comments to the Author” section, enter your conflict of interest statement in the “Confidential to Editor” section, and submit your "Accept" recommendation.

Reviewer #1: All comments have been addressed

Reviewer #2: All comments have been addressed

2. Is the manuscript technically sound, and do the data support the conclusions?

Reviewer #1: Partly

Reviewer #2: Yes

3. Has the statistical analysis been performed appropriately and rigorously? 

Reviewer #1: Yes

Reviewer #2: Yes

4. Have the authors made all data underlying the findings in their manuscript fully available?

Reviewer #1: Yes

Reviewer #2: Yes

5. Is the manuscript presented in an intelligible fashion and written in standard English?

Reviewer #1: Yes

Reviewer #2: Yes

6. Review Comments to the Author

Reviewer #1: In statistical analysis it is stated that “odds ratio was measured with a relative risk”. Relative risk is a measure for prospective studies and odds ratio is a measure for retrospective studies. To clear the confusion it maybe more appropriate to say something like “odds ratio was used as a risk measure”.

Also when I checked the frequencies of tables, just for one comparison a table has an expected count less than 5, so instead of using Fisher’s exact test for all tables, it would be better to use continuity corrected chi-square test if all cells have an expected count higher than 5 and also have an observed count less than 25. If all cells have an expected count higher than 5 and also all cells have an observed count higher than 25, it would be appropriate to use Pearson’s chi-square test for determining the significance.

Another issue, what was the name of the post-hoc procedure used after one-way ANOVA for pairwise comparisons. Please mention its name in statistical analysis section.

Authors stated that they corrected the footnote of Table 1. But instead of SEM, it still remains as SD.

As the odds ratios are corrected, at lines 215-217 the statement should be also corrected, instead of saying the heterozygous and 240bp/240bp homozygous are associated with risk of substance abuse, it would be correct to say these two genotypes are protective for substance abuse. Because their odds ratios are lower than 1. This means the reference homozygous 120/120 is more likely to become substance abuser.

At lines 232-233, the statement should be heterozygous and 240bp/240bp homozygous have less risk to become a cannabis abuser (odds ratios are less than 1), 120bp/120bp homozygous is more likely to become cannabis abuser.

At lines 257-259, it is stated that post-hoc analysis was done for COMT, as seen in table 5, one-way analysis resulted as ns. If the omnibus test is not significant no need to perform a post-hoc analysis. Also, the statement in the same manner at lines 318-319 is unnecessary as omnibus test result is p>0.05.

Reviewer #2: The authors appear to have made all the requested changes. However, there are other minor changes to consider.

Line 53 – “very high percentage (93%0 of addicted subjects in Bangladesh are male”. Remove 0 after 93% and close the parenthesis.

Line 166 – “are less educated (18% illiterate)”. Where is this information found in Table 1? Better to add this information to the table or change it to secondary educated.

Lines 167-169 – “case of COMT polymorphism, the graduate and employed subjects are less likely to be addicted than the secondary educated and unemployed subjects.” Was that information supposed to appear in the table? The question is being raised, as there is no information about the polymorphism of COMT.

Line 191 – “suicidal attempt (12.4%)”. Is this information in accordance with table 2?

Line 205 – “COMT 158Val/Met” - COMT Val158Met

Line 228 - (e.g methamphetamine) => (e.g., methamphetamine)

Lines 229-230 - (e.g methamphetamine abuser) => (e.g., methamphetamine abuser)

Line 335 – “addicted patients were illiterate and unemployed compared to the control subjects” – In line 335, the authors state that a significantly higher percentage of dependent patients was illiterate and point out that this information is present in Table 1, however the information cited is not included.

Lines 339-340 – “and the study of Hasam and Mushahid [31] and Islam et al. [32]” => and the studies of Hasam and Mushahid [31] and Islam et al. [32]

Line 345 – “Although only 9.4% abusers” – In the results and in table 2, the value is 9.5%.

Lines 348-349 – “significant relationship was found between the heterozygous genotype of COMT polymorphism (Table 3)”. It could indicate the relationship with what, for example, “significant relationship was found between the heterozygous genotype of COMT polymorphism and substance abuse (Table 3)”.

Lines 362-371 – The authors state that “significant relationship was found between both the derived genotype of DRD4 tandem duplication and risk for substance abuse where according to the odds ratio, both heterozygous (120bp/240bp) (OR=0.37, 95% CI=0.172-0.826) and homozygous (240 bp/240 bp) (OR=0.43, 95% CI=0.193-0.937) derived variants showed association with decreased risk of substance abuse” and “the longer allele (240bp/240bp) of the tandem duplication played a protective role in substance abuse, and according to a previous study by Kereszturi et al., 2007, the shorter allele (120bp/120bp) is found to be the risk allele in novelty seeking and other neuropsychiatric behavior”. Justifying why the longer allele is not associated with substance abuse. However, on lines 371-376 the explanation seems to me to be about how the longer allele is associated with substance abuse.

Line 370 – “by Kereszturi et al., 2007” – standardize citations.

The authors could write a conclusion for the article.

7. PLOS authors have the option to publish the peer review history of their article (what does this mean?). If published, this will include your full peer review and any attached files.

Reviewer #1: No

Reviewer #2: No

---

## [Author Response · Author response to Decision Letter 1]

2 Jan 2021

A point-by-point reply to Reviewers comments:    

Reviewer # 1:

Comment: In statistical analysis, it is stated that the "odds ratio was measured with a relative risk." Relative risk is a measure for prospective studies, and the odds ratio is a measure for retrospective studies. To clear the confusion, it may be more appropriate to say something like "odds ratio was used as a risk measure."

Author’s Reply: Yes, we agree with the reviewer. In statistical analysis, the statement “odds ratio was measured with a relative risk at 95% confidence intervals” was replaced by “odds ratio was used as a risk measure at 95% confidence intervals” (Line: 155).

Comment: Also, when I checked the frequencies of tables, just for one comparison, a table has an expected count less than 5, so instead of using Fisher's exact test for all tables, it would be better to use continuity corrected chi-square test if all cells have an expected count higher than 5 and also have an observed count less than 25. If all cells have an expected count higher than 5, and also all cells have an observed count higher than 25, it would be appropriate to use Pearson's chi-square test for determining the significance.

Author’s Reply: Authors agreed with the reviewers about this statistical procedure. As the Fischer exacts test is a much more acceptable test for genetic polymorphism analysis by most scientists, authors also used that test. However, upon receiving the reviewer’s comments, the authors reanalyzed the data as recommended and found similar associations as with Fischer Exact tests. The authors updated the information in the manuscript and modified the methodology.

Comment: Another issue was the name of the post-hoc procedure used after one-way ANOVA for pairwise comparisons. Please mention its name in the statistical analysis section.

Author’s Reply: The post-hoc analysis used after one-way ANOVA was Tukey's Test for post-hoc analysis, which is mentioned in the revised manuscript (Lines: 159-160).

Comment: Authors stated that they corrected the footnote of Table 1. But instead of SEM, it still remains as SD. 

Author’s Reply: Thank you for indicating this mistake. Accordingly, the footnote of Table 1 was corrected from SD to SEM (Line: 183). 

Comment: As the odds ratios are corrected, at lines 215-217, the statement should also be corrected, instead of saying the heterozygous and 240bp/240bp homozygous are associated with risk of substance abuse, it would be correct to say these two genotypes are protective for substance abuse. Because their odds ratios are lower than 1. This means the reference homozygous 120/120 is more likely to become a substance abuser. 

Author’s Reply: Yes, we agree with the reviewer. As the odds ratios are lower than 1, in the revised manuscript, the statement “both the heterozygous and homozygous derived variants were found significantly associated with risk of substance abuse compared to the control subjects” was replaced by “Both heterozygous (120bp/240bp) and homozygous (240 bp/240 bp) derived variants were found to show a protective role for substance abuse compared to the control subjects which was statistically significant" (Lines: 217-218). 

Comment: At lines 232-233, the statement should be heterozygous and 240bp/240bp homozygous have less risk to become a cannabis abuser (odds ratios are less than 1), 120bp/120bp homozygous is more likely to become cannabis abuser. 

Author’s Reply: According to the suggestion of the reviewer, as the odds ratios are less than 1, the statement “In the case of DRD4 polymorphism, both the heterozygous and homozygous derived variants of cannabis abusers were significantly associated with the risk of cannabis abuse (p<0.01)" was replaced by "In the case of DRD4 polymorphism, it was found that both the heterozygous(120bp/240bp) and homozygous(240bp/240bp) derived variants have less risk to become cannabis abusers which were statistically significant (p<0.01)" in the revised manuscript (Lines: 233- 235)

Comment: At lines 257-259, it is stated that post-hoc analysis was done for COMT, as seen in table 5, one-way analysis resulted as ns. If the omnibus test is not significant no need to perform a post-hoc analysis. Also, the statement in the same manner at lines 318-319 is unnecessary as the omnibus test result is p>0.05. 

Author’s Reply: Very Good Suggestion. In the revised manuscript, the statement "To compare pairwise differences among the age of onset of three different genotypic groups of COMT polymorphism post-hoc analysis was made, no significant difference was found” was deleted accordingly. Also, the statement “Further, after post-hoc analysis, there was no pairwise difference among these genotypic groups of DRD4 polymorphism” at line 318-319 was also deleted accordingly.

Reviewer #2: 

Comment: Line 53 – “very high percentage (93%0 of addicted subjects in Bangladesh are male”. Remove 0 after 93% and close the parenthesis. 

Author’s Reply: In the revised manuscript at line 53, 0 is removed after 93%, and the parenthesis was closed.

Comment: Line 166 – “are less educated (18% illiterate)”. Where is this information found in Table 1? Better to add this information to the table or change it to secondary educated. 

Author’s Reply: Thank you for indicating the mistake. During the previous revision of the manuscript, due to the unavailability of illiterate subjects in the control group and multivariate analysis, we combined drug-addicted illiterate subjects (18%) with secondary educated subjects and compared them with the graduate subjects. After combining the percentage of secondary educated subjects became 84.7%. By mistake, the information was not changed at line 166. In the revised manuscript, this information is corrected in line 166.

Comment: Lines 167-169 – “case of COMT polymorphism, the graduate and employed subjects are less likely to be addicted than the secondary educated and unemployed subjects.” Was that information supposed to appear in the table? The question is being raised, as there is no information about the polymorphism of COMT. 

Author’s Reply: We performed multiple logistic regression using all the variables and found that in the case of COMT polymorphism, the graduate and employed subjects are less likely to be addicted than the secondary educated and unemployed subjects. We did not mention this association in Table 1, as we have shown all the genetic associations in individual Tables. 

Comment: Line 191 – “suicidal attempt (12.4%)”. Is this information in accordance with table 2.

Author’s Reply: As according to the study, 5.9% of addicted subjects made a suicidal attempt; in the revised manuscript, the information "suicidal attempt (12.4%)" was changed into "suicidal attempt (5.9%)" according to Table 2.

Comment: Line 205 – “COMT 158Val/Met” - COMT Val158Met.

Author’s Reply: According to the reviewer’s suggestion, “COMT 158Val/Met” is rewritten as “COMT Val158Met” at line 205.

Comment: Line 228 - (e.g methamphetamine) => (e.g., methamphetamine) 

Author’s Reply: According to the reviewer’s suggestion, (e.g. methamphetamine) is rewritten (e.g., methamphetamine) at line 229 in the revised manuscript.

Comment: Lines 229-230 - (e.g methamphetamine abuser) => (e.g., methamphetamine abuser) 

Author’s Reply: According to the reviewer’s suggestion, (e.g methamphetamine abuser) is rewritten (e.g., methamphetamine abuser) at lines 230-231 in the revised manuscript. 

Comment: Line 335 – “addicted patients were illiterate and unemployed compared to the control subjects” – In line 335, the authors state that a significantly higher percentage of dependent patients was illiterate and point out that this information is present in Table 1, however the information cited is not included. 

Author’s Reply: As the information about Educational status given in Table 1 did not include illiterate subjects, in the revised manuscript, the statement "a significantly higher percentage of dependent patients was illiterate" was replaced by "a significantly higher percentage of addicted patients were less educated” (Line 334), as in Table 1 it is mentioned that 84.7% addicted subjects had secondary education. 

Comment: Lines 339-340 – “and the study of Hasam and Mushahid [31] and Islam et al. [32]” => and the studies of Hasam and Mushahid [31] and Islam et al. 

Author’s Reply: According to the reviewer’s suggestion, “and the study of Hasam and Mushahid [31] and Islam et al. [32]” is rewritten as “and the studies of Hasam and Mushahid [31] and Islam et al.”

Comment: Line 345 – “Although only 9.4% abusers” – In the results and in table 2, the value is 9.5%. 

Author’s Reply: Yes, we agree with the reviewer. According to Table 2, the percentage of addicted subjects with relapse behavior was 9.5%; at line 344 in the revised manuscript, this information is rewritten as “although only 9.5% abusers”.

Comment: Lines 348-349 – “significant relationship was found between the heterozygous genotype of COMT polymorphism (Table 3)”. It could indicate the relationship with what, for example, “significant relationship was found between the heterozygous genotype of COMT polymorphism and substance abuse (Table 3)”. 

Author’s Reply: According to the suggestion of the reviewer, the statement at lines 347-348 in the revised manuscript is rewritten as “a significant relationship was found between the heterozygous genotype of COMT polymorphism and substance abuse."

Comment: Lines 362-371 – The authors state that “significant relationship was found between both the derived genotype of DRD4 tandem duplication and risk for substance abuse where according to the odds ratio, both heterozygous (120bp/240bp) (OR=0.37, 95% CI=0.172-0.826) and homozygous (240 bp/240 bp) (OR=0.43, 95% CI=0.193-0.937) derived variants showed association with decreased risk of substance abuse” and “the longer allele (240bp/240bp) of the tandem duplication played a protective role in substance abuse, and according to a previous study by Kereszturi et al., 2007, the shorter allele (120bp/120bp) is found to be the risk allele in novelty seeking and other neuropsychiatric behavior". Justifying why the longer allele is not associated with substance abuse. However, on lines 371-376, the explanation seems to me to be about how the longer allele is associated with substance abuse. 

Author’s Reply: Yes, we agree with the reviewer that the explanation given in lines 371-376 did not indicate how the longer allele is associated with less substance abuse risk. As the longer allele of DRD4 120bp tandem duplication played a protective role in Bangladeshi substance abused subjects, a better explanation behind the reason is given at lines 370-376 in the revised manuscript.

Comment: Line 370 – “by Kereszturi et al., 2007” – standardize citations. 

Author’s Reply: According to the reviewer's suggestion, the citation is standardized in the revised manuscript (Line 369).

Comment: The authors could write a conclusion for the article. 

Author’s Reply: According to the reviewer's suggestion, a conclusion is added in the revised manuscript (Lines: 448-450).

---

## [Decision Letter · Decision Letter 2]

20 Jan 2021

Catechol-O-methyltransferase and dopamine receptor D4 gene variants: possible association with substance abuse in Bangladeshi male

PONE-D-20-16794R2

Dear Dr. Kabir,

We’re pleased to inform you that your manuscript has been judged scientifically suitable for publication and will be formally accepted for publication once it meets all outstanding technical requirements.

Kind regards,

Zhicheng Lin, Ph.D.

Academic Editor

PLOS ONE

Additional Editor Comments (optional):

Reviewers' comments:

Reviewer's Responses to Questions

**Comments to the Author**

1. If the authors have adequately addressed your comments raised in a previous round of review and you feel that this manuscript is now acceptable for publication, you may indicate that here to bypass the “Comments to the Author” section, enter your conflict of interest statement in the “Confidential to Editor” section, and submit your "Accept" recommendation.

Reviewer #1: All comments have been addressed

Reviewer #2: All comments have been addressed

2. Is the manuscript technically sound, and do the data support the conclusions?

Reviewer #1: Yes

Reviewer #2: Yes

3. Has the statistical analysis been performed appropriately and rigorously? 

Reviewer #1: Yes

Reviewer #2: Yes

4. Have the authors made all data underlying the findings in their manuscript fully available?

Reviewer #1: Yes

Reviewer #2: Yes

5. Is the manuscript presented in an intelligible fashion and written in standard English?

Reviewer #1: Yes

Reviewer #2: Yes

6. Review Comments to the Author

Reviewer #1: (No Response)

Reviewer #2: (No Response)

7. PLOS authors have the option to publish the peer review history of their article (what does this mean?). If published, this will include your full peer review and any attached files.

Reviewer #1: No

Reviewer #2: No

---

## [Editor Report · Acceptance letter]

25 Jan 2021

PONE-D-20-16794R2 

Catechol-O-methyltransferase and dopamine receptor D4 gene variants: possible association with substance abuse in Bangladeshi male 

Dear Dr. Kabir:

I'm pleased to inform you that your manuscript has been deemed suitable for publication in PLOS ONE. Congratulations! Your manuscript is now with our production department. 

Kind regards, 

on behalf of

Professor Zhicheng Lin 

Academic Editor

PLOS ONE